# Low-index mesoscopic surface reconstructions of Au surfaces using Bayesian force fields

Cameron J. Owen [1,2] ✉, Yu Xie [2], Anders Johansson [2], Lixin Sun[2,4] & Boris Kozinsky[2,3] ✉

Metal surfaces have long been known to reconstruct, significantly influencing their structural and catalytic properties. Many key mechanistic aspects of these subtle transformations remain poorly understood due to limitations of previous simulation approaches. Using active learning of Bayesian machine-learned force fields trained from ab initio calculations, we enable large-scale molecular dynamics simulations to describe the thermodynamics and time evolution of the low-index mesoscopic surface reconstructions of Au (e.g., the Au(111)-'Herringbone,' Au(110)-(1 × 2)-'Missing-Row,' and Au(100)-'Quasi-Hex-agonal' reconstructions). This capability yields direct atomistic understanding of the dynamic emergence of these surface states from their initial facets, providing previously inaccessible information such as nucleation kinetics and a complete mechanistic interpretation of reconstruction under the effects of strain and local deviations from the original stoichiometry. We successfully reproduce previous experimental observations of reconstructions on pristine surfaces and provide quantitative predictions of the emergence of spinodal decomposition and localized reconstruction in response to strain at non-ideal stoichiometries. A unified mechanistic explanation is presented of the kinetic and thermodynamic factors driving surface reconstruction. Furthermore, we study surface reconstructions on Au nanoparticles, where characteristic (111) and (100) reconstructions spontaneously appear on a variety of high-symmetry particle morphologies.

Accurate description of surfaces and their dynamic evolution is an important task in materials modeling, as the resulting structures strongly influence device performance and stability. Examples of these effects range from interfacial reactions dictating transport and degradation at electrolyte-electrode interfaces[1–4], surface morphologies affecting tribology of mechanical devices[5], to the structure-performance relationship of heterogeneous catalysts affecting turn-over rates for chemical conversion processes[6]. Until explicit consideration of interfacial responses to environmental stimuli (e.g., applied strain, temperature, or adsorbates) using experimental or computational methods is achieved with atomic resolution, interpretation of these phenomena and subsequent material design tasks will remain largely intractable.

Of these dynamic phenomena, surface reconstructions have proven nominally difficult to capture using existing methods, where ab initio techniques like density functional theory (DFT) cannot simulate appropriate length- or time-scales, and classical techniques (e.g., empirical force-fields (FFs)) do not exhibit the required accuracy[7].

[1]Department of Chemistry and Chemical Biology, Harvard University, Cambridge, MA, USA. [2]John A. Paulson School of Engineering and Applied Sciences, Harvard University, Cambridge, MA, USA. [3]Robert Bosch LLC Research and Technology Center, Watertown, MA, USA. [4]Present address: Microsoft Research, Cambridge, UK. ✉e-mail: cowen@g.harvard.edu; bkoz@seas.harvard.edu

Coupling these limitations to insufficient temporal and spatial resolution of experimental techniques means that the atomistic mechanisms and nucleation kinetics of surface reconstruction remain unknown.

Gold (Au) has garnered particular attention in this regard due to the propensity of each of its low-index surfaces to reconstruct under inert conditions. The Au(111)[8–10], Au(110)[11–13], and Au(100)[14,15] surfaces each exhibit interesting reconstructions[16]. Specifically, Au surfaces, and more generally those of other late transition metals used in heterogeneous catalysts, have been shown to exhibit different activities only after reconstruction has occurred, where changes in preferential adsorption of reactants[17,18], as well as alloying behaviors are caused by the change in morphology of the surface[19]. In addition to reconstructions, Au surfaces exhibit a spinodal decomposition, as observed using experimental STM by Schuster et al.[20]. Charcteristic labyrinth patterns were seen with an Au adatom gas on the Au(111) surface between coverages of 0.4 and 0.9 monolayers (ML), with the dominant lengthscales of the agglomerated islands of only a few nanometers. Hence, reproducing and understanding these reconstruction processes from atomistic simulations, and the resulting differences in material properties is important specifically for Au as well as other widely-used host metals in a variety of applications[21].

Such mesoscopic modeling challenges naturally lend themselves to be solved using molecular dynamics (MD) simulations or ab initio methods, where the former can appropriately simulate the large time- and length-scales necessary for these reconstructions, while the latter exhibit high accuracy in describing atomic environments of diverse coordination but are limited by high computational cost. Corroborating the need for increased accuracy at appropriate scale is the fact that MD simulations driven by either classical FFs or ab initio methods have been unable to capture Au surface reconstructions. While several classical embedded atom method (EAM) FFs are readily available for Au, all of which having been trained using different physical properties, none of them are able to capture the low-index reconstructions of Au. Instead, explicitly guided MD simulations have been performed[22], wherein the atomic structure of the reconstructed surface is directly built or the MD simulation is manually adjusted to prompt reconstruction (e.g., applied shear) and simulated using these potentials[23], yielding no atomistic insight into the nucleation kinetics or mechanisms of the reconstruction processes. The need for guided structure manipulation in MD simulations began with the inability of Frenkel-Kontorova (FK) models to predict the stability of the Au(111)-'Herringbone' reconstruction[24], followed by modifications to same FK models that were able to stabilize the facet but not directly observe the transformation[25], extending to other EAM potentials that also required explicit construction of the target facet[26].

Recent advances of machine learning in FF development have shown promise for direct simulation of these mesoscopic phenomena, as a flexible model can be learned directly from ab initio training data and enable MD simulations at high computational efficiency on par with classical potentials. These models demonstrate high accuracy in both in- and out-of-domain modeling tasks[7,27–30]. Moreover, ML force fields (MLFFs) have become easier to train fully autonomously, and allow production MD simulations to reach experimentally relevant scale (e.g., 0.5 trillion atoms for H/Pt heterogeneous catalytic reactions[31]). MLFF models are trained on ab initio training data, either using density functional theory (DFT) for periodic systems or quantum-chemistry methods for molecules, from which atomic forces, total energies, and stresses can be employed as 'ground-truth' labels. As a result, MLFFs exhibit accuracy close to that of their ab initio training data, and when coupled with their flexible forms, they permit robust calculation of material properties at increasingly appropriate scales for comparison to experimental observations[29,31]. With respect to Au, a DeepMD MLFF was recently used for investigation of the

Au(111) 'Herringbone' reconstruction, but instead of exploring its emergence, explicit guidance was again employed to study the reconstructed facet by directly building it prior to the simulation, which limits the predictive analysis of the transformation behavior[32]. However, we do find it apparent here to note that the work using the DeepMD potential took advantage of DFT training labels at the PBE level, which when used to study the resultant 'Herringbone' reconstruction (e.g., DeepMD using training labels at the PBE-level), agreement was obtained with respect to experimental observations for various perturbations of the subsurface atomic layers, and strain induced changes in periodicity. There also exists a wealth of previous static DFT investigations, which when taken together conclude that the choice of exchange-correlation functional, albeit important, does not influence the stability assessments of the reconstructed facets relative to the pristine systems when using generalized gradient approximations like PBE[32,33]. Issues may arise when using small periodicity 'approximates' (supercell representations, as defined in ref. 33) to predict the surface energy of the mesoscopic reconstructions as identified in ref. 33 for Pt(100), but this is distinct from the simulation tasks presented here, since we simulate the entire length-scale required for the full surface reconstruction of each facet. Moreover, the work from ref. 33 specifically notes that even the use of small periodicity approximates for the Au(100) at the GGA level with something like PBE gives reasonable agreement to experimental observations.

This argument is similar to another critical component of the work completed here, in that not only are small approximates (supercell representations) used to train a MLFF, but the descriptors used to represent atomic environments in the FLARE code are strictly local, where information is not propagated beyond the cutoff distance. Symmetry-breaking phenomena, such as charge density waves, which arise from delicate interactions among the various orbitals, are governed by long-range fundamental electronic structure features[34], which may be difficult to capture using strictly local representations. However, our non-trivial finding is that the emergence of such long-range patterns of reconstruction, due to strain and electronic effects, can in fact be described in quantitative agreement with experiment by a model that is able to sufficiently accurately capture only short-range quantum interactions.

Here, we accomplish the goal of direct and unbiased observation of low-index Au surface reconstructions by employing the FLARE code[7,35,36] to construct a MLFF from ab initio training data that is able to capture, without explicit guidance and guessing of structure, the dynamic surface reconstructions of each of the low-index facets of Au, as well as on nanoparticles (NPs). A summary of the workflow is provided in Fig. 1, wherein a single MLFF was trained for Au, validated, and then deployed in large-scale ML-MD simulations to study surface reconstruction of slabs and NPs under a variety of boundary conditions. As provided in Fig. 1, small atomic unit-cells of Au bulk, surfaces, and nanoparticles (1a) are fed into the FLARE framework (1b) to efficiently trained a FLARE MLFF 'on-the-fly', from which unbiased ML-MD simulations can be performed (1c) to uncover atomistic understanding of surface reconstructions and their nucleation (1d) directly from first principles. With these simulations, we provide direct and unbiased insights into a full mechanistic understanding of surface reconstruction and nucleation kinetics under which Au surface reconstructions readily occur. These observations are important for improved understanding of catalyst synthesis, pretreatment, and to potentially control reactivity of these systems. By establishing a rigorous protocol by which surface reconstructions can be studied directly from ab initio training data using MLFFs, we open the possibility of computational investigations of the mechanisms, kinetics, and thermodynamics of a wide range of surface reconstruction phenomena and interpretation of experimental results in surface science. Our work aims to address fundamental surface science questions. (1) How does the kinetic

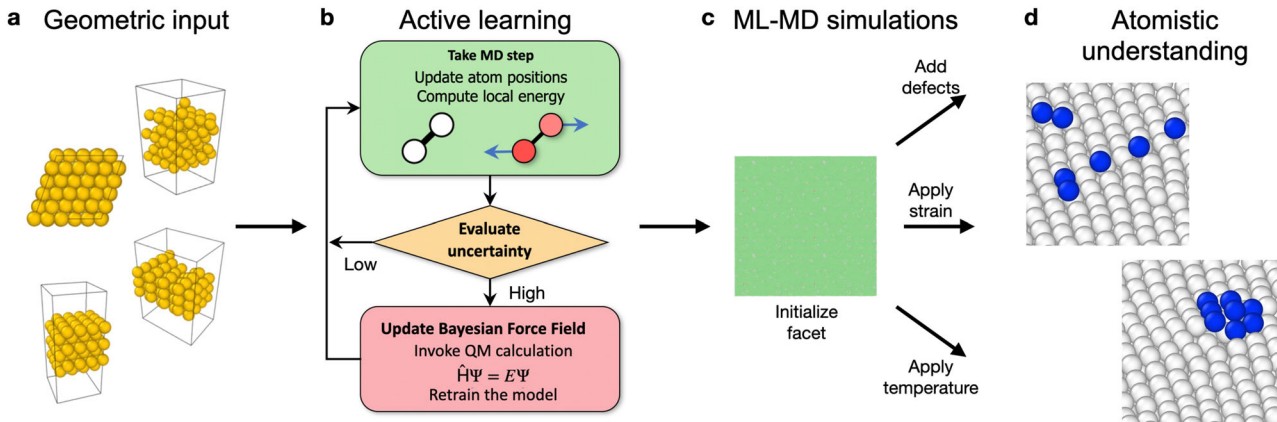

**Fig. 1 | Schematic of the data generation and simulation workflow. a** DFT-sized input geometries for active learning in FLARE. **b** Active learning in FLARE, as described in more detail in refs. 7 and[36]; adapted with permission (CC BY 4.0). **c** ML-MD simulations performed in LAMMPS for various facets with defects, mechanical strains, and temperatures applied. **d** Nucleation kinetics and mechanisms underlying the transition towards the reconstructed phases are made available to determine the stoichiometric and mechanical regimes wherein each surface is reconstructed, and what is the resulting surface geometry.

emergence of reconstruction depend on the initial surface structure, specifically applied strain, concentration of adatoms, and presence of edges? (2) Are surface reconstructions thermodynamically favored under a wide range of these factors? (3) Are short-range interatomic interactions sufficient to capture intricate large-scale reconstruction patterns observed in experiment? (4) How does the reconstruction phenomenology on flat surfaces transfer to the evolution of nanoparticle facets? The results presented below provide insight into these important questions for Au facets, and ultimately yield benchmarks to be confirmed by experimental surface science techniques in the coming years, in particular the explicit dependence of strain and surface stoichiometry on Au reconstruction of flat terraces and nanoparticles, as well as the onset of a spinodal decomposition that would appear following surface cleaning, e.g., sputtering, at low temperatures.

## Results

The procedures for active learning, MLFF training, and validation are provided in the "Methods" Section, and the results of each are presented in the Supplementary Information. Briefly, active learning only resulted in 2965 calls to DFT, across the 7 structures considered over 13.2 ns of total simulation time, all collected within 1775.1 h (≈74 days) of serial wall time. The active learning procedure is represented schematically in Fig. 1, where atomic structures are input to the initially empty sparse Gaussian process model, which is trained iteratively via MD simulations. In practice, by running all active learning trajectories in parallel, all data was collected only over the span of 1 week of CPU wall time. Excellent agreement is observed across all validation targets considered, which established preliminary trust in the MLFF to interpolate between each of the relevant Au surfaces and bulk environments. The MLFF was then employed to study the low-index surfaces and several Au NPs in large-scale ML-MD simulations, discussed sequentially below.

### Au(111)-'Herringbone' reconstruction

Au is the only FCC-metal whose (111) surface is observed to reconstruct at room temperature[8–10]. The resulting 'Herringbone' periodicity $(22 \times \sqrt{3})$ has been experimentally imaged using scanning tunneling microscopy (STM), which can be found in refs. 37,38.

To provide insight into the nucleation kinetics and mechanisms driving reconstruction, we deployed our MLFF in large-scale ML-MD simulations with various surface stoichiometries, defined by the concentration of vacancies or adatoms, introduced in or on the surface atomic layer, relative to the pristine facet, of the Au(111) surface as

input and evaluated the propensity of each surface to reconstruct. The simulation procedure is described in detail in the "Methods" Section and is also represented schematically in Fig. 1, where the trained FLARE model is used to describe various Au facets and nanoparticles under stimuli with unbiased MD to yield atomistic understanding. To account for various factors that could plausibly influence the onset of reconstruction, as provided in Fig. 1c, we defined three Simulation Tasks: (1) the stoichiometric Au(111) facet at 300 K across a range of mechanical strains, (2) the same facet and applied strains but with adatoms or vacancies randomly introduced across the entire concentration range (from 0.05 to 0.95 ML), and (3) heating and subsequent quenching of the stoichiometric surface.

Simulation Tasks 1 and 2 were considered at 300 K, as this temperature is appropriate for comparison to most high-vacuum surface science characterizations of Au(111) single crystals[9]. At this temperature, the coupled effects of stoichiometry and mechanical strain (both anisotropic and isotropic, as explained in the "Methods" Section) were explicitly probed, since these perturbations have been shown to influence the periodicity of reconstruction[32]. However, no experimental or computational insight is currently available into their influence on the apparent nucleation kinetics and mechanisms underlying reconstruction. Lastly, we evaluated the stoichiometric Au(111) surface under annealing conditions starting from 300 K to temperatures between 400 and 800 K in increments of 100 K, with a heating rate of 20 K per ns that was previously used for Au by Zeni et al.[39]. The results for these tasks are summarized in Figs. 2 and 3, respectively, and the complete set of results are provided in Suppl. Note 2.

### Stoichiometric Au(111)

Beginning with Task 1, we observe that reconstruction is strongly dependent on applied strain, as shown in Fig. 2a, b, where only isotropic tensile strain above 1.5% along both [1 − 10] and [−1 − 12] is able to nucleate the transition within the simulation time of 10 ns at 300 K. A snapshot of the final reconstructed facet after 10 ns of ML-MD is shown in Fig. 2a for the case where 2.0% isotropic tensile strain is applied, which leads to nucleation within 200 ps of simulation time. This time-scale is different than for application of 1.5% isotropic tensile strain, which takes ≈1.2 ns to nucleate. These results are discussed in more detail below. The pristine Au(111) simulation cell consists of 39,600 atoms, and obtained an average performance of 55 ns per day using 4 A100 GPUs.

In order to monitor the presence of reconstruction over the course of each simulation, the polyhedral template matching (PTM)

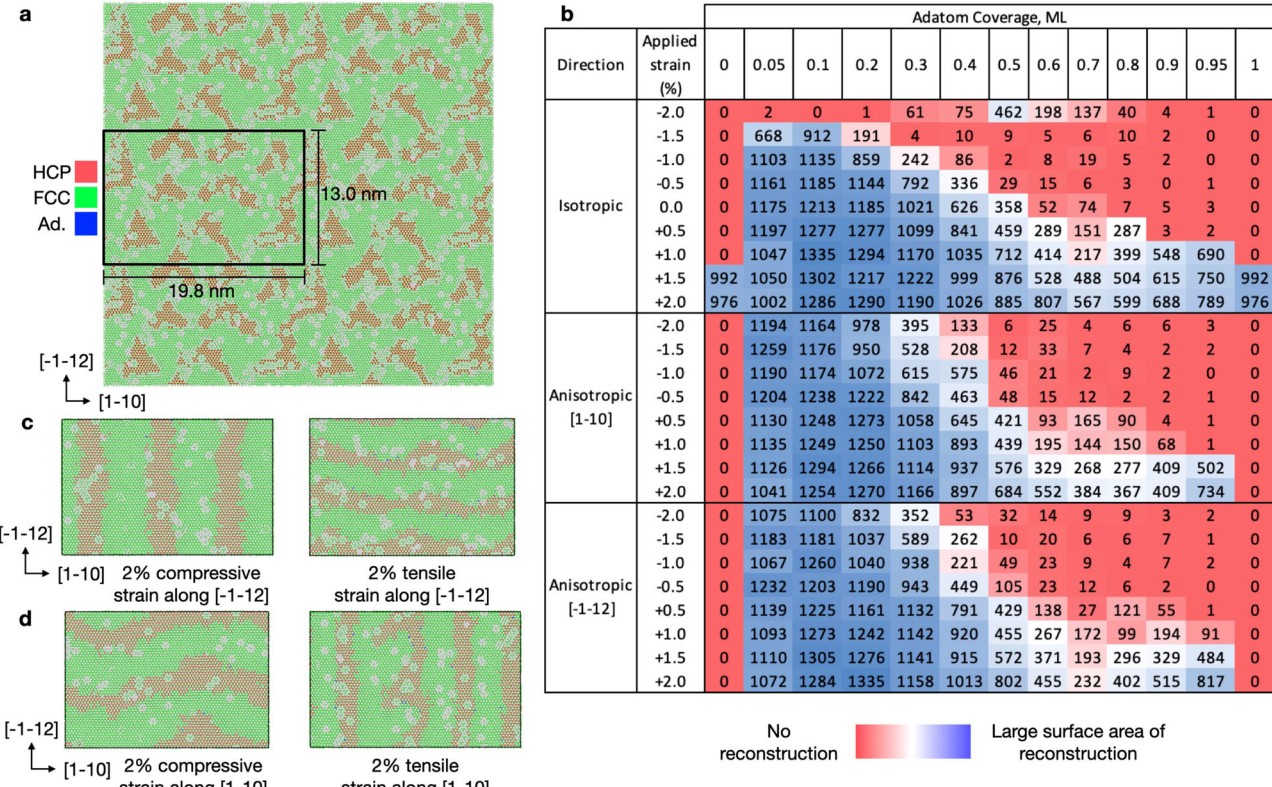

**Fig. 2 | Appearance of Au(111) reconstructions under strain and changes in surface stoichiometry. a** Final snapshot of the Au(111) surface after 10 ns of simulation time under 2% isotropic tensile strain. Atoms are colored using the Polyhedral Template Matching (PTM)[40] method in Ovito[41], where green denotes FCC, red denotes HCP, blue denotes BCC, and white denotes lack of symmetry of a given atomic environment. **b** The kinetic regimes in which facile reconstruction occurs as a function of strain and surface stoichiometry are observed. Values on the top x-axis correspond to the surface monolayer (ML) coverage of adatoms and the y-axis provides strain directions and relative amounts in percent. The values represent the number of HCP atomic environments present at the end of each simulation, obtained from PTM. The blue-white-red heat map is included to provide a guide to the eye for when reconstruction fully occurs (blue) or does not nucleate (red) within the allotted simulation time. **c, d** Snapshots from four separate ML-MD simulations with opposite directions of applied strain, as defined in Simulation Task 2, demonstrating the effect of mechanical stimulus on the periodicity of reconstruction and how the applied strain results in periodicities of reconstruction that are orthogonally related.

method[40] was employed using Ovito[41], where atomic environments are distinguished by the distribution of angles of neighboring atoms. From these results we can conclude that the stoichiometric Au(111) surface is resistant to reconstruction, with only high amounts of isotropic tensile strain prompting nucleation. However, we do point the reader to Table 1, which provides the relative energies of the reconstructed and pristine facets as a function of isotropic strain, as calculated by the MLFF. These results are in excellent agreement with those previously reported in the literature[42], where the striped Au(111)-'Herringbone' reconstructed facet is more stable than the pristine facet at 0.0% strain, which is reproduced by our MLFF.

From these results, the reconstructed facet is still the enthalpic minimum unless significant isotropic compression is applied. This is consistent with the trend shown by Fig. 2b, but also indicates that strain affects both the kinetic barriers to reconstruction and the thermodynamic driving force. However, we use caution in making strong conclusions from this small set of static calculations, but reiterate that the trends in energy for the non-strained facet reflect those previously observed in ref. 42, and serve as a source of additional validation for the MLFF.

### Defective Au(111)

We then considered the inclusion of adatoms and vacancies on Au(111) to yield a range of 'defective' surfaces to understand the effect of surface stoichiometry on the mechanisms and nucleation kinetics of reconstruction. These results are provided in Fig. 2a–d. As opposed to the simulations presented in Task 1, reconstruction is observed regardless of strain, unless 2.0% isotropic compression is applied. This is explained in terms of the high effective barrier for adatom incorporation into the compressed surface atomic layer with the enthalpic minimum at 2.0% isotropic compression being the (111) facet, as is supported by the enthalpic information in Table 1. For context, the 'Herringbone' reconstruction requires that an additional two atoms be included in each $(22 \times \sqrt{3})$ unit cell of the surface, changing the total number of surface atoms from 44 to 46 with this periodicity. More interesting is that a change in periodic pattern is observed as a function of anisotropic strain, as shown in Fig. 2c, d. This is corroborated by the resulting direction of the reconstruction being rotated by 90° when comparing compression or tension between the two lattice vectors. If anisotropic mechanical strain is applied along an individual lattice vector but is varied from compression to tension, however, the reconstruction also rotates by 90°. This suggests that the direction of strain is critical in determining the resulting periodic patterning of the reconstructed domains.

These findings of the dependence of the periodic boundary conditions on the 'Herringbone' reconstruction are consistent with experiments conducted by Schaff et al.[43], and corroborate the results provided by our model. The energetic preference for 'striped' domains was also observed via applied strain to the 'Herringbone' construction considered using the DeePMD potential[32], but they did not consider evolution towards this lowest energy structure directly from the perfect facet, neglecting the nucleation kinetics presented here.

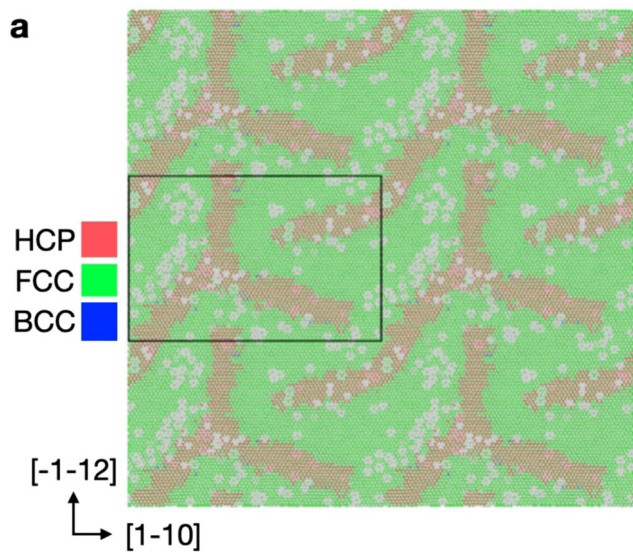

HCP
FCC
BCC

[-1-12]
[1-10]

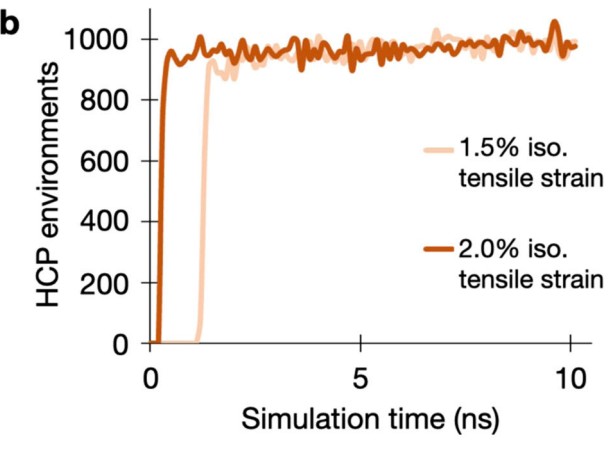

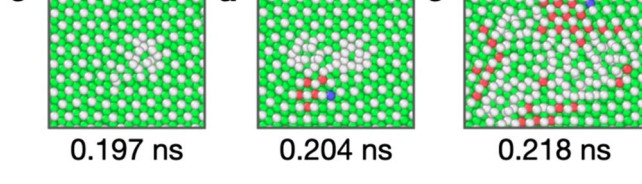

0.197 ns   0.204 ns   0.218 ns

**Fig. 3 | Appearance of Au(111) reconstructions under temperature and nucleation kinetics. a** Final snapshot of the 500 K simulation in Task 3 following the 20 K ns⁻¹ quench back down to 300 K. **b** Number of HCP atomic environments as a function of simulation time in Task 1, from which the total time-scale of reconstruction can be determined. **c**–**e** Snapshots of the nucleation center(s) predicted by the MLFF for reconstruction of the stoichiometric Au(111) surface under 2.0% isotropic tensile strain.

**Table 1 | Energy comparison of the striped Au(111)-'Herringbone' reconstructed facet versus pristine Au(111) at different values of isotropic strain**

| | Striped Au(111)-'Herringbone' | | Au(111) | |
|---|---|---|---|---|
| Isotropic strain | $E$ per atom (eV) | $\gamma$ (eV nm⁻²) | $E$ per atom (eV) | $\gamma$ (eV nm⁻²) |
| −2.0 | −3.1895 | 5.0375 | -3.1830 | 4.7713 |
| 0.0 | −3.1932 | 4.1937 | -3.1850 | 4.3182 |
| +2.0 | −3.1859 | 5.2549 | -3.1761 | 5.2923 |

periodicities and patterns can be directly observed. We explain these observations in more detail below.

At the lowest concentration considered (0.05 ML), this directly corresponds to the 'Herringbone' stoichiometry, which explains the agreement in the periodicity of reconstruction. Exceeding 0.5 ML of adatoms into the 'vacancy-rich' domain, we observe a continued reduction of the spatial extent of the reconstructed domains, until 0.95 ML adatoms are introduced (or equivalently 0.05 ML of vacancies). Ultimately, reconstruction is only observed on the 0.95 ML surface when tensile strain of 1.0% or higher is employed, along either lattice direction, or isotropically. This result differs slightly from the pristine Au(111) surface in Task 1 without defects, where reconstruction only occurred under isotropic tensile strain of 1.5% and 2.0%, so the inclusion of vacancies allows for reconstruction to appear under a broader range of boundary conditions. In effect, the presence of defects lowers the nucleation free energy barrier for reconstruction, as evidenced by the extent of reconstruction provided here.

These observations hold immediate experimental relevance, since pretreatment of Au single crystals typically include a 'cleaning' procedure, e.g., where Ar gas is used to sputter the surface, ultimately creating vacancies, or an ion beam is used to deposit Au atoms onto the sample as prepared. Hence, our MLFF allows for atomistic understanding of the effects of surface stoichiometry and mechanical stimulus on the resulting mesoscopic reconstruction, which can help inform and explain the effects of these experimental surface science procedures and underlay the sensitivity of surface reconstruction kinetics and thermodynamics to the surface conditions.

### Influence of temperature on stoichiometric Au(111)

Experimental studies of Au single crystals typically follow cleaning procedures with annealing and quenching protocols to allow the system to minimize its energy. Hence, we also consider the effect of temperature applied to the perfect Au(111) surface, the results of which are provided in Fig. 3. The 'Herringbone' reconstruction was only observed after 500 K was reached during annealing, a snapshot of which is shown in Fig. 3a. Reconstruction is observed at all temperatures above 500 K until surface pre-melting, which we observe with our potential at 800 K. Surface pre-melting and loss of order at a lower temperature than experiment (≈950 K[44]) can be explained by the underestimation of the Au force constant and lattice parameter via employment of the PBE exchange correlation functional to generate the ab initio training set, as has also been observed for NP melting point determinations made by Zeni et al.[39]. We leave a systematic investigation of stability and kinetics of reconstruction as a function of temperature to a future work.

### Nucleation and kinetics of reconstruction

An advantage of our ML-MD approach is that it can be used to directly probe the nucleation time-scales of reconstruction across the wide set of stimuli. An example of this is shown in Fig. 3b for the two simulations which exhibited reconstruction in Task 1. Hence, we can determine the time-scale of reconstruction, as well as the influence of applied tensile strain. Here, the fully reconstructed facet appears within 400 ps for 2.0% strain, and within 1.5 ns for 1.5% strain. Visualization of the atomic

To generalize our investigation of the defective Au(111) surface, we also considered the entire stoichiometric range from 0.05 ML to 0.95 ML of adatoms, which effectively covers both adatom- and vacancy-rich domains. We observed a clear effect on the periodic length scale of the reconstructed HCP domains across the range of surface concentrations. Specifically, reconstructed domains exhibit the largest periodicity when the adatom concentration is low, yielding domains closest to those observed experimentally (e.g., $22 \times \sqrt{3}$), whereas increasing the concentration of adatoms to 0.5 ML and higher results in the reconstructed domains being much shorter in length-scale. The complete set of simulation snapshots as a function of surface stoichimetry is provided in Suppl. Figs. 5–15, where these different

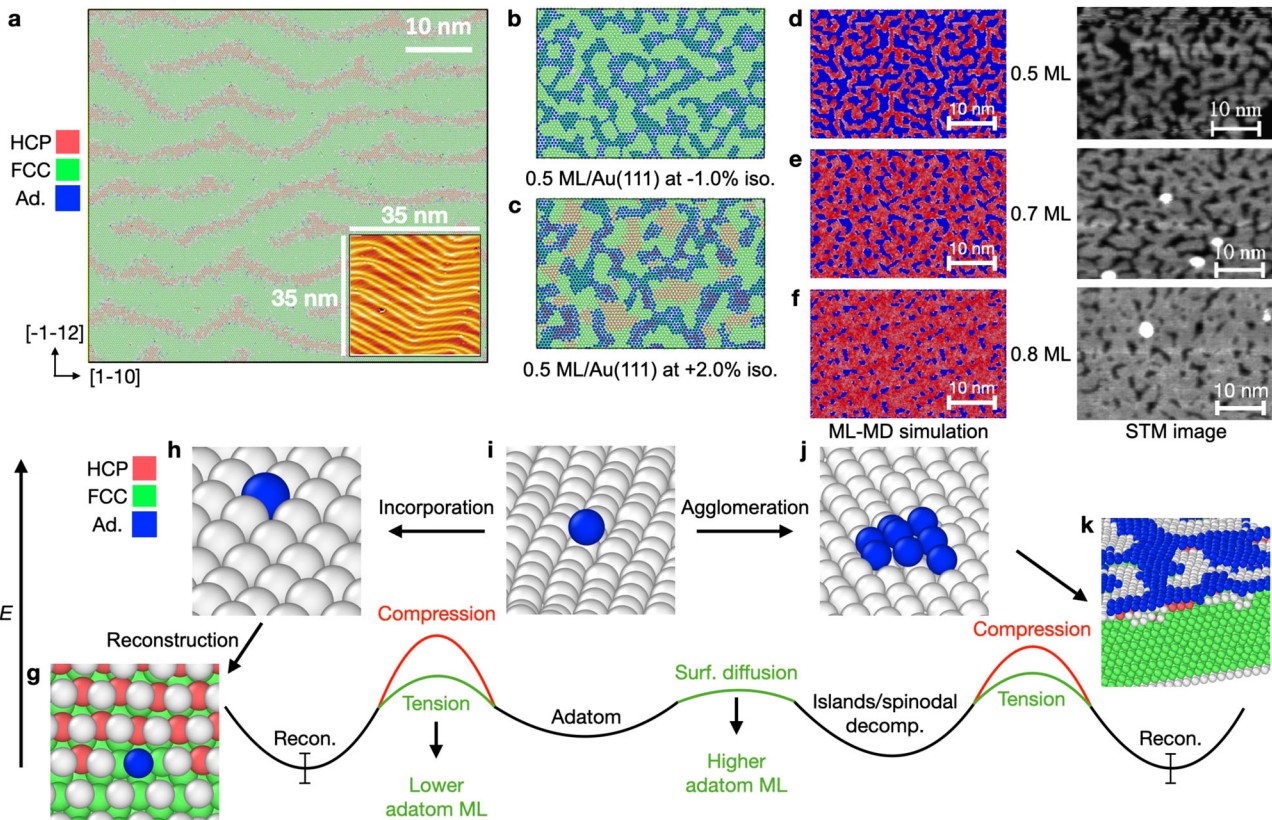

**Fig. 4 | Full mechanistic understanding of Au(111) reconstructions. a** Snapshot of the Au(111)-'Herringbone' reconstruction from a simulation containing ≈ 330, 000 atoms, where the atoms are colored using the PTM method in Ovito. Again, red denotes HCP and green denotes FCC packing of the surface atoms, and the orthogonal surface vectors are provided. An experimental STM image is provided as an inset; adapted with permission from ref. 38 (Copyright 2023 American Chemical Society). **b** Spinodal decomposition for 0.5 ML adatoms on the Au(111) surface under compressive and **c** tensile strain. Adatoms are colored blue, and PTM is employed to distinguish if and where reconstruction occurs, using the same color scheme as in **a**. **d** Snapshots for 0.5 ML, **e** 0.7 ML, and **f** 0.8 ML adatoms on Au(111) with no applied mechanical strain from both ML-MD and scanning tunneling microscopy (STM) images, adapted with permission from ref. 20 (Copyright 2003

American Physical Society). **g–k** Qualitative schematic detailing the energy landscape accessible to adatoms as in **i** at the onset of these simulations, where they can incorporate into the surface as in **g** and **h**, prompting reconstruction, the barrier of which and thermodynamic preference of which are both influenced by applied mechanical strain. On the other hand, the adatoms can diffuse with essentially no barrier to quickly form islands by spinodal decomposition as in **j**, where the edge atoms can then incorporate into the subsurface to prompt reconstruction as in **k**, the barriers and thermodynamic minima of which are again both influenced by applied mechanical strain. The rightmost image is cut to show the subsurface composition of atomic environments, primarily FCC, but HCP in the immediate subsurface atomic layer, denoting the presence of reconstruction near the Au-adatom island.

environments at the nucleation point is also provided for the simulation employing isotropic tensile strain of 2.0%, where growth of the reconstructed phase occurs over the course of ≈200 ps, the snapshots of which are included in Fig. 3c. From these snapshots, clusters of atoms differing from the original FCC symmetry can be identified, which then begin to grow over a period of ≈ 20 ps. Only once the growth has reached a critical surface area, do HCP domains begin to appear. We do not permit an extended focus on the study of nucleation phenomena here, but note that this method can be used to make such determinations, which may be the subject of future investigations. As for Simulation Tasks 2 and 3, nucleation snapshots are not provided, since nucleation was observed to happen immediately (i.e., during structural optimization) across almost all simulations, indicating a negligible nucleation barrier. Hence, our MLFF is able to provide direct atomic insights into the nucleation phenomena that ultimately result in these mesoscopic reconstructions, which have not been previously possible to obtain.

### Surface phase stability and kinetic regimes

An important take-away from this work is that Bayesian MLFFs are key enablers to survey surface reconstructions without bias, including kinetic and thermodynamic aspects. Importantly, one can now use

direct ML-MD to determine surface phase stability with respect to several variables. An important observation from these simulations, corroborating this statement, is the appearance of spinodal decomposition patterns on the Au(111) surface at a range of concentrations of adatoms. In addition to the full 'Herringbone' reconstruction, which was captured at the scale of ≈330,000 atoms in Fig. 4a, whereas Fig. 4b, c provide simulation snapshots at varying adatom coverages and applied strains. In Fig. 4b, spinodal decomposition is observed, with the adatoms shown in blue, and the other surface and subsurface atoms colored by PTM in OVITO. These simulations were both initialized by randomly placing adatoms, which then quickly diffuse and agglomerate into nanoscale interdigitated networks of islands within 1 ns that remain stationary throughout the entire subsequent simulation. A more complete set of these results for spinodal decomposition is provided in Suppl. Fig. 4, specifically for the Au(111) surface across 0.1–0.9 ML of adatoms, without applied mechanical strain. The structure and nanometer length scale of spinodal decomposition structures shown in Fig. 4c agree remarkably well with experimental STM work by Schuster et al.[20]. Our simulations not only quantitatively capture experimentally observed patterns but also provide mechanistic and time scale details of their formation and response to strain.

By visualizing these simulations with PTM, we can make conclusions about the appearance of reconstruction in accordance with the spinodal decomposition patterns that appear. The 'Herringbone' reconstruction primarily occurs in the exposed terrace, as shown by the red patches in Fig. 4c, where small patches of HCP envriouments also appear underneath the islands. Conceivably, the atoms at the edges of the agglomerated islands are responsible for nucleating the reconstruction, and the number of these environments and their ability to drive such processes depends on the total adatom coverage, as does the fraction of the exposed surface that is able to reconstruct.

These mesoscale considerations also explain the previously discussed results in Fig. 2b, mainly with respect to the reduction of reconstruction amounts at increasing adatom coverage. We use Fig. 4d to illustrate these details in the context of the role of adatoms. As illustrated in the center of this schematic, randomly placed adatoms are used to initialize our simulations, which can either incorporate into the surface layer, or diffuse along the surface and agglomerate to form small islands via spinodal decomposition. The preference to go towards incorporation or agglomeration is influenced by the local stoichiometry of the surface, as well as the applied strain. If tension is applied and there is a dilute coverage of adatoms, the barrier for incorporation into the subsurface is reduced, allowing for the system to quickly approach the 'Herringbone' stoichiometry, and reduce the high energy of the adatom from its initially low-coordination environment. On the other hand, if adatom coverage is high, wherein island on the surface are not at the right stoichiometry and have limited responses to tension, or compressive strain in applied, there is a higher preference for the adatoms to agglomerate, which also reduces the high energy of the adatom from its initially low-coordination environment.

Hence, we have demonstrated the ability of our MLFF and its use in large-scale ML-MD simulations to study the previously inaccessible coupled effects of mechanical strain and surface adatom/vacancy concentration on the emergence of surface reconstruction. From these simulations, both nucleation kinetics and fully atomistic mechanisms were uncovered, allowing for improved design of materials important for catalysis and other applications.

## Au(100)-'quasi-hexagonal' reconstruction

Additionally, we considered the Au(100) facet which, likewise to Au(111), has been shown to reconstruct under inert experimental conditions[45,46]. Hence, a similar approach was employed, where the effect of strain was first investigated for reconstruction of the stoichiometric facet, the results of which are summarized in Fig. 5. We make an explicit note here that the color scheme used for the analysis of each surface is unique, so the color assignments should not be used across facets of different symmetry. For example, the HCP environments which are denoted using the color red and the atoms in Fig. 5 should not be interpreted in the same fashion, and these differences are made explicit in the caption of each figure.

Reconstruction of stoichiometric Au(100) surface is observed in Fig. 5a, and occurs almost immediately following equilibration during ML-MD at 300 K. Stripes appear, denoting appearance of the 'quasi-hexagonal' reconstruction, with (N × 5) periodicity as inferred by the atomic heights of the surface layer. This explicit periodicity is not immediately obvious via visualization of Fig. 5a, but is clear in Fig. 5f when isotropic compression of 1.5% is applied to the cell. Here, a single reconstructed domain can be observed after 10 ns of ML-MD that exhibits a periodicity of (5 × 20), as has been determined experimentally, and is shown in 5b from ref. 47. For nearly all of the simulations that fully reconstruct, orthogonal stripes of these close-packed reconstructed domains appear along the [01 − 1] and [011] lattice directions, which are qualitatively consistent with the high-resolution STM results presented in[48].

An important observation is also made with respect to this reconstruction being localized to the surface atomic layer, as shown in Fig. 5c, where only the surface atoms are shown with low-coordination change from their simple cubic termination to 'quasi-hexagonal'. This snapshot is taken from the same simulation frame as in Fig. 5a, and is on the same color-scale, but the surface atomic-layer has been removed to directly visualize the subsurface atoms. Atoms directly underneath the surface layer from Fig. 5a retain their simple-cubic packing, while atoms that become exposed due to the appearance of 'vacancy-pits' reconstruct in the same 'quasi-hexagonal' fashion as the removed surface layer. Hence, the coordination number of atoms on the Au(100) surface is a vital factor that controls the appearance of reconstruction, without the need for inclusion of adatoms, vacancies, strain, or temperature, pointing to the absence of a nucleation barrier, unlike the pristine Au(111) surface.

Likewise to the Au(111) surface, however, high values of compressive strain kinetically hinder reconstruction. Tensile strain, on the other hand, results in a drastic change of the surface morphology, leading to the appearance of 'vacancy-pits' on the surface, denoted by blue splotches (i.e., removal of surface layer density, since the reconstructed domains have higher packing density). These are observed in both tensile and compressive simulations, with both anisotropic and isotropic directions of strain, however their surface area increases drastically when the simulation cell undergoes tensile deformation. The effect of strain on this reconstruction has not been as rigorously investigated for the Au(111) surface, so these results need to be validated by future experimental queries.

## Au(110)-'Missing-Row' reconstruction

To round off the set of Au low-index surfaces, we also considered Au(110), which has also been shown to reconstruct at room temperature under inert conditions to yield the (1 × 2) 'Missing-Row' reconstruction[11–13]. Other periodicities have also been observed, namely the (1 × 3)[49], and (1 × 4)[50] reconstructions. With this context in mind, this surface was studied in a similar fashion to Au(111), where the coupled effects of strain and stoichiometry were considered. The influence of defects was deemed necessary to initialize the ML-MD simulations for this system since the stoichiometric Au(110) surface requires a substantial rearrangement of the atoms to provide the correct atomic ordering of the reconstructed phase. This is due to the high kinetic barriers to form the 'Missing-Row' Au(110) surface from the pristine facet, confirmed by long time-scale simulations at 300 K that did not exhibit reconstruction on the order of 10 ns. These results are provided in Fig. 6.

Ultimately, the 'Missing-Row' reconstruction was observed, as shown in Fig. 6a, b, denoted by bright red streaks coupled with bright blue streaks and a periodicity of (1 × 2). This simulation was initialized via random removal of 0.5 ML of surface atoms, and then allowed to evolve at 300 K. We also considered the entire concentration range of 0.1–0.9 ML of adatoms, snapshots of which are provided in Fig. 6c. Likewise to the Au(100) system, the atoms are colored by their heights in the surface-normal direction. Briefly, the 'ridge' features at greater heights (in bright red) of the reconstruction for the stoichiometric facet (0.5 ML) are more easily observed when tensile strain is applied to the system, whereas compression leads to less obvious patterns. An assessment of nucleation and timescales is not considered for this surface, given the enormous number of 'defect' environments. Hence, defining a time-scale for nucleation is thus not straightforward and left for a future investigation.

## Surface reconstruction on nanoparticles

Lastly, we used the same MLFF model to study reconstructions on a variety of surface structures present on NPs. These systems include multiple facets, combined with edges, vertices, and non-FCC bulk environments (e.g., HCP or icosahedral). Notably, surface reconstructions of NPs provide a more difficult simulation task than the preceding extended terraces due to the variety of environments and the

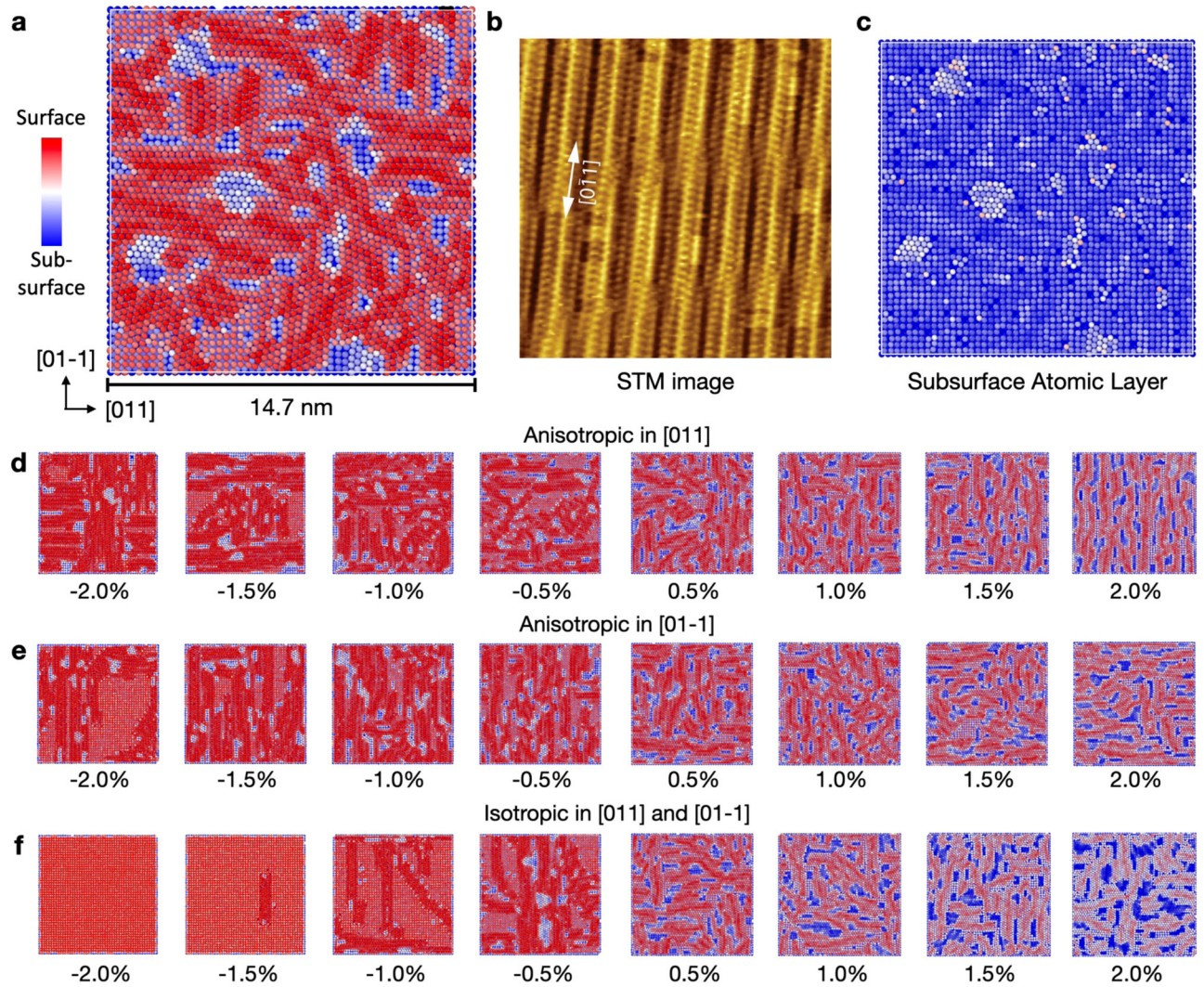

**Fig. 5 | Au(100)-'quasi-hexagonal' reconstruction. a** Final snapshot of the Au(100) surface following 10 ns ML-MD at 300 K with no strain applied. Atoms are colored by their height in the surface-normal direction, where blue denotes atoms in the subsurface atomic layer, and red denotes atoms in the surface atomic layer. **b** Experimental STM image of the reconstructed Au(100) surface, adapted with permission from ref. 64 (CC BY 4.0). **c** Snapshot of the same surface in **a**, but the surface layer is removed to view the subsurface atoms and their registry with the bulk (i.e., they retain simple-cubic packing). **d**–**f** Final snapshots of the strained Au(100) surface simulations along [01 − 1] and [011] lattice directions starting with the stoichiometric surface.

large number of atoms needed to represent the NP structures. To limit the breadth of structural motifs to consider in this regard, three high-symmetry NP shapes were chosen: an icosahedron, cuboctahedron, and ino-truncated decahedron, each of which exhibits (111), (100), or a combination of these surface facets. All particles considered here are free-standing and all NP simulations were kept at 300 K to allow for comparison with the low-index flat surfaces. Figure 7 provides snapshots of the initial particle shapes following structural minimization within LAMMPS, and the final structures of 104,223-atom NPs after 10 ns of ML-MD simulation, the conclusions from which are discussed below. This is another demonstration of the capabilities of Bayesian MLFF simulations, compared to earlier studies, which were limited by the aforementioned inabilities of previous interatomic potentials in correctly describing nanoparticle structures and their surface dynamics simultaneously.

**Exclusive (111) faceting on an icoshedra**
Starting with the icosahedron, as shown in Fig. 7a–c, all exposed facets are (111), separated by edges and internal bulk domains of HCP-packing, the latter seen as red atoms in Fig. 7b,c. Following 10 ns of ML-MD, as shown in Fig. 7b, c, however, each facet contains streaks of HCP

domains, indicative of 'Herringbone'-like surface reconstructions. This suggests that even in the absence of explicitly created 'defective' atomic environments, like those observed for the flat periodic Au(111) surfaces, the (111) surfaces of the NP are able to reconstruct readily at 300 K, since the appearance of the HCP domains in the subsurface of each (111) facet of the particle denotes reconstruction of the surface layer of atoms. Upon closer inspection of the ML-MD trajectories that connect the left and right snapshots of Fig. 7a–c, we can conclude that this is due to the presence of edge and vertex atomic environment of the NP, which exhibit small displacements and rounding during simulation, indicative of atoms being incorporated into the surface planes of each (111) facet to induce reconstruction. This is an important observation, as these domains would likely influence the adsorption motifs of reactive species in catalytic settings. This hypothesis will be tested in a follow-up investigation via augmentation of the MLFF training set to also include adsorbates.

**Cuboctahedra with (111) and (100) faceting**
We then shift our attention to the cuboctahedral particle displayed in Fig. 7d–f. Here, we make observations in line with the (111) facets on extended terraces and the preceding icosahedron, which are shown to

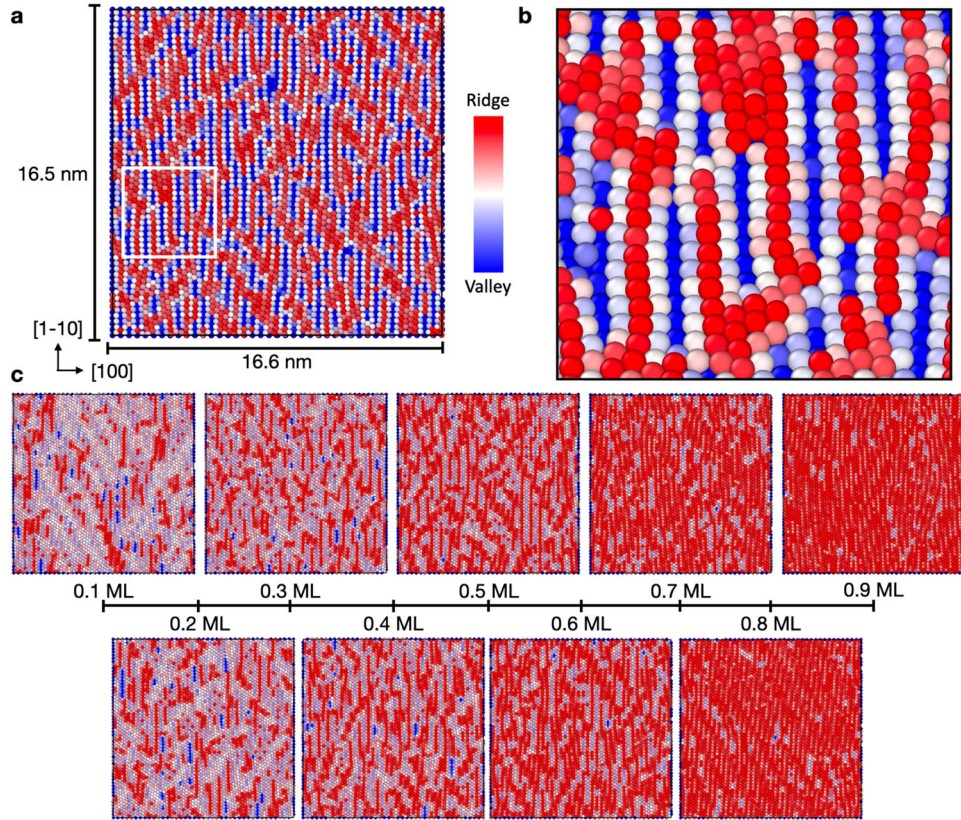

**Fig. 6 | Au(110)-'missing-row' reconstruction. a** Final snapshot of the Au(110) surface following 10 ns of production ML-MD using a random deletion of 0.5 ML of the surface atoms. Atoms are colored by height using the Ovito software[41], where red denotes a greater *z*-coordinate and blue denotes a lower value. Local domains present on the surface have reconstructed to yield the `Missing-Row'-like reconstructions, denoted by bright red streaks along the [1$\bar{1}$0] lattice vector of the cell. **b** Zoomed-in snapshot of the (1×2) reconstructed domain in **a**. **c** Final snapshots of the simulations under different surface stoichiometries ranging from 0.1 to 0.9 ML coverage of adatoms via random inclusion.

reconstruct in a similar manner. Interestingly, the (111) facet of the cuboctahedron exhibits reconstructed domains immediately after geometric relaxation of the NP. This observation partially confirms our edge-atom nucleation hypothesis from the case for the icosahedron, where edge atoms incorporated into the bordering (111) surface planes to promote reconstruction. This can be directly observed in the left snapshot of Fig. 7f, where HCP domains emanate from each edge of the facet, and slight distortions of the atomic positions of the edge environments can be seen. These partially reconstructed domains then aggregate once the system evolves during ML-MD, to yield one large HCP domain and a small artifact near one of the edges, as observed in the final snapshot of the system.

Dissimilar from the icosahedron, the cuboctahedron also exhibits (100) facets that border the (111) facets on each edge. Unlike the (111) facets, which reconstruct immediately, the (100) surfaces reconstruct only after the system evolves during production ML-MD at 300 K. Similar observations are made with respect to the Au(100) extended terrace discussed previously, where streaks of hexagonal, close-packed atoms appear, shown in Fig. 7e. The 'quasi-hexagonal' reconstruction is again confirmed to be limited to only the surface atoms, as the subsurface atomic layer remains in registry with the FCC-stacking of the bulk.

**Ino-truncated decahedra with (111) and (100) faceting**

Lastly, we consider the ino-truncated decahedral NP, which exhibits both (111) and (100) facets like the cuboctrahedon, but with edge environments that are different than the cuboctahedron due to particle symmetry. In the cuboctahedron, the (111) facets are only surrounded by (100) edges, whereas (111) facets on the ino-decahedron share edges with both (111) and (100) facets. Similarly, however, bright red reconstructed domains are immediately observed on all (111) facets on the particle following structural minimization. Again, this is explained as originating from the edge atomic environments along the (100) facet, which relax into the (111) surface. This observation corroborates what was seen for the cuboctahedral particle, where these (111)-(100) edge environments immediately nucleate the 'Herringbone'-like reconstruction on the (111) facet. These reconstructions also retain the symmetry of the particle, being five-fold along the high-symmetry axis, and do not evolve away from this state during production ML-MD as shown in Fig. 7i. Likewise to the cuboctahedron, the (100) facets remain largely unaffected following minimization, exhibiting their initial simple-cubic packing, but once the system evolves in production ML-MD, each reconstructs to yield 'quasi-hexagonal' domains of (N × 5).

## Discussion

An interesting and nontrivial finding from this work is that the observed mesoscopic reconstructions are several tens of nanometers in length, but can be predicted by our ML-MD simulations based on MLFF which employs a descriptor cutoff of 6 Å. To illustrate further why this finding is important, consider the fact that the Au(111)-'Her-ringbone' reconstruction has a periodicity of (22 × $\sqrt{3}$), which has a length-scale of ≈65 Å, meaning that the atomic descriptors that are trained from ab initio data cannot observe the full periodicity of this reconstruction but are able to simulate it regardless of this spatial limitation. This is also corroborated by the fact that the ab initio training set does not contain the reconstructed surface. This then implies that the observed mesoscopic reconstructions result from

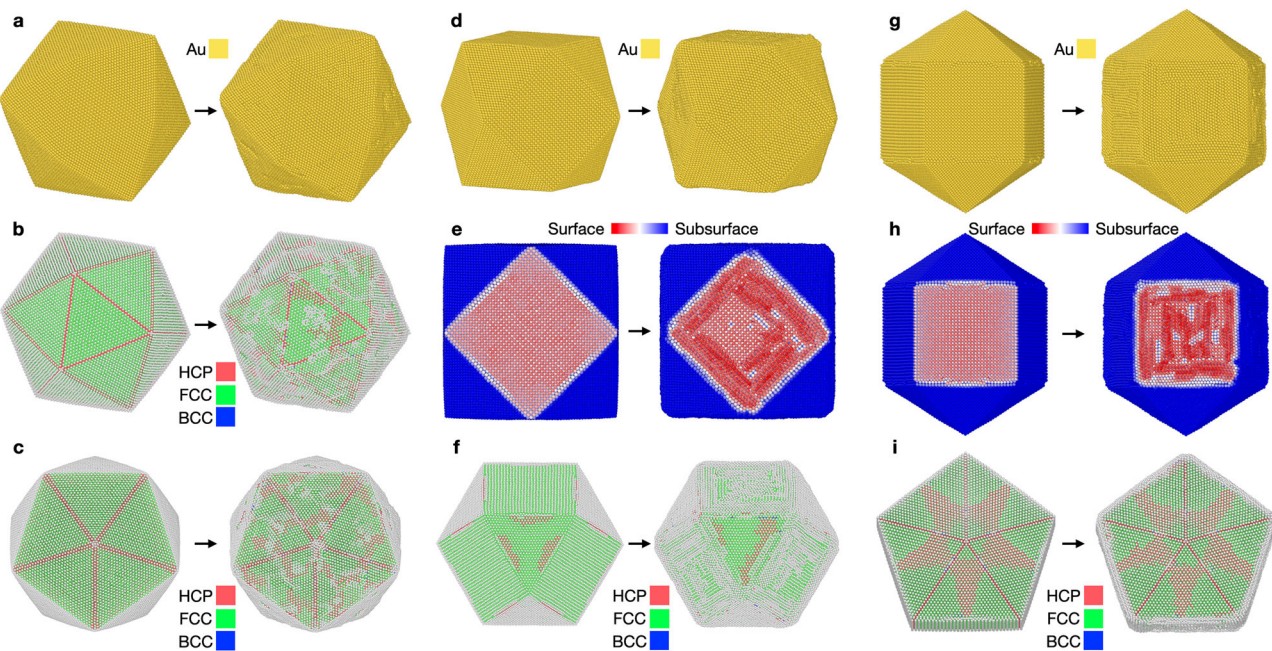

**Fig. 7 | Snapshots of the initial (0 ns) and final (10 ns) geometries of three Au-NP shapes. a** Au$_{104223}$ icoshedron, where all surfaces exhibit (111) faceting. **b**, **c** Atoms are colored by their environment using the PTM method provided in the Ovito software[41], where red denotes HCP- and green denotes FCC-packing. **d**–**f** Au$_{104223}$ cuboctahedron, where the surface area is divided between (100) and (111) facets. **e** Atoms are colored by their height using Ovito or using the PTM scheme in **f**. **g**–**i** Au$_{104223}$ ino-truncated decahedron, which also exhibits both (100) and (111) surfaces like the cuboctahedron. **h**, **i** employ the same coloring schemes as the cuboctahedron.

short-range, local interactions resulting in relaxations of the atomic structure. This result is partially corroborated by previous results by Frenkel-Kontorova[24], where they developed a ball-and-spring force field for the treatment of surface reconstructions. That effort sought to explain (111) surface reconstructions using a series of one-dimensional spring interactions between the surface atoms and the underlying subsurface and bulk atoms. Ultimately, this local potential was able to explain the instability of (111) reconstructions of FCC-metals platinum, iridium, and aluminum, but ultimately failed in its predictions for gold, due to its limited accuracy in capturing small energy differences determining surface behavior.

This study provides a determination of quantitative time-scales and mechanistic insight into nucleation of each of the surface reconstructions of Au(111), Au(100), Au(110) surfaces and nanoparticles under the coupled effects of stoichiometry, temperature, and strain. Ultimately, defects in the surface environments are found to sensitively affect surface reconstruction in a non-trivial fashion, where adatoms, by their incorporation into the surface, promote reconstruction regardless of strain, while vacancies allow for slight deviations from the observed behavior of the pristine facets. In partially covered surfaces, we observe systematic formation of spinodal decomposition leading to the formation of nanoscale island networks. We are able to quantitatively explain previous observations from experimental microscopy in this regime and further predict that reconstruction is primarily spatially localized in the base surface between the islands. For the pristine surfaces, however, tensile strain was found to be largely influential in allowing for facile reconstruction. Moreover, we demonstrate the presence of a nucleation barrier for reconstruction of the unstrained pristine Au(111) surface, which is not the case for Au(100) and Au(110) or NPs. With respect to the latter, edge environments along bordering facets of different symmetry in nanoparticles were found to nucleate reconstruction, as was observed for the cubotrahedral and ino-truncated decahedral particles. Enabling this analysis is the development of a Bayesian force field from ab initio data that is able to reveal the atomistic mechanisms of surface

reconstructions using large-scale molecular dynamics simulations. We find that only short-range many-body interactions are sufficient to accurately produce a wide array of even long-range reconstruction patterns. Hence, this work establishes the ability and utility of machine learning driven dynamics simulations for directly capturing large length-scale and time-scale dynamics of metal surface reconstructions with minimal human input, thus providing direct insight into subtle interactions and surface restructuring mechanisms. This work enables future investigations of scientifically and technologically important effects of temperature, strain, defects and molecular adsorbents on surface structure evolution, opening possibilities of rational design of heterogeneous catalytic and nano-scale devices.

## Methods
### Active Learning with FLARE
The Bayesian active learning module within the Fast Learning of Atomistic Rare Events (FLARE) code is described in detail in refs. 7,51. FLARE is open-source and available at the following repository: https://github.com/mir-group/flare. Briefly, atomic environments are described using the atomic cluster expansion from Drautz (ACE)[52] Sparse Gaussian process kernel regression is employed to compare atomic descriptors, providing an inherent mechanism to quantify uncertainties of these environments, which are used during the active learning simulation to select ab initio training data 'on-the-fly.' Thus, atomic environments are only added to the sparse set of the Gaussian process if their relative uncertainties are higher than a predefined threshold, which is set by the user, allowing for fast sampling and collection of high-fidelity DFT frames. Reference 36 contains a schematic of the FLARE active learning framework employed here, where each simulation is initialized using a given structure and computing atomic properties using the Sparse GP model.

Since the sparse set of the GP is initially empty, all atomic environments are determined to be high-uncertainty at the start of each active learning trajectory, and DFT is called. The threshold for calling DFT was set to 0.01 in most cases. This value for the threshold

represents the uncertainty of a given atom relative to the mean uncertainty of all atoms in the system. Consequently, a call was made to the Vienna ab initio Simulation Package (VASP, v5.4.4)[53–56], where DFT training labels were generated for the structure. The DFT parameters employed for each system are provided in the next section. A separate threshold was employed to determine how many atoms should be added into the sparse-set following each DFT call, and was set to be ≈20% of the DFT threshold. The magnitudes of these thresholds are important parameters that can be tuned by the user, as a smaller sparse-set threshold will add more atomic environments for each DFT call, which could result in the SGP being dominated by non-unique environments. Conversely, setting this value too high (with the limit being a 1:1 match with the DFT threshold) will result in only a few, or even just a single atom being added to the sparse set for every call to DFT. When the thresholds are chosen correctly, this method is much more efficient than ab initio MD, where a DFT calculation is required at every time-step ($\Delta t = 5$ fs in this case). The atomic positions in the initial frame of each simulation were randomly perturbed by 0.01 Å.

Once high-uncertainty atomic environments were selected from the DFT frame and added to the sparse-set, the SGP was then re-mapped onto a lower-dimensional surrogate model and the system was allowed to continue in the ML-MD simulation. Atom positions were then updated with respect to forces using the LAMMPS Nosé-Hoover NVT ensemble until another atomic environment was deemed as high uncertainty by the SGP. The FLARE hyperparameters (signal variance $\sigma$, energy noise ($\sigma_E$), force noise ($\sigma_F$), and stress noise ($\sigma_S$)) were optimized during each active learning training simulation up until the 20$^{th}$ DFT call. Each optimization step was allowed to run for a total of 200 gradient descent steps, which was sufficient for hyperparameter convergence. The priors assigned to each hyperparameter were set to empirical values observed previously for bulk Au FLARE B2 models[30], specifically: 3.0, 0.001$N_{atoms}$ (eV), 0.1 (eV Å$^{-1}$), and 0.001 (eV Å$^{-3}$), respectively. The exact values of these priors are not crucial in the FLARE framework, since hyperparameter optimization is performed and they then become dominated by training data once a sufficient number of DFT calls and subsequent optimization steps have been made. Each of the low-index Au-surfaces were considered during parallel active learning simulations, as well as bulk Au with and without tensile and compressive strains and nanoparticles of various sizes. A summary of these systems and their active learning results are provided in Suppl. Note 1.

Following completion of the set of parallel active learning trajectories, all of the DFT frames were collected, from which the final FLARE MLFF was trained. MLFF training followed the same selection procedure for atomic environments as described above, where atomic environments were selected based on their relative uncertainties, without performing any molecular dynamics, referred to as 'offline-learning.' Subsequently, we also rescaled the energy noise hyperparameter of the resulting MLFF to account for multiple systems of different sizes and structures being included in the training set. This hyperparameter describes the noise level of the energy labels of the training dataset, and can get trapped in the local minima during the optimization which affects the model accuracy. Therefore, we rescaled noise hyperparameter of energy labels to 1 meV per atom, which did not influence the force or stress hyperparameters.

For each of the parallel active learning trajectories, as well as the final offline-training, the B2 ACE descriptor was employed, maintaining consistency in notation with Drautz[52]. By taking the atomic descriptor to the second power, this accounts for 'effective' 5-body interactions within each descriptor, which are sufficiently complex for describing Au with high-accuracy[30]. To explore the effect of static model parameters, 5 frames from a Au(111) active learning trajectory were used as a training set, the remaining frames as the test set, to perform a grid-search. Ultimately, a radial basis ($n_{max}$) of 8, angular basis ($\ell_{max}$) of 3, and a cutoff radius ($r_{cut}$) of 6.0 Å were found to

increase the log-marginal likelihood to a maximum value while reducing the energy, force, and stress errors to their respective minima. These results are also consistent with those determined previously for Au from the TM23 dataset[30]. Parity and the associated errors between the final MLFF and DFT on energy, force, and stress predictions are provided in Suppl. Fig. 3, along with several other validation protocols.

### Density functional theory

The Vienna ab initio Simulation Package (VASP, v5.4.4)[53–56] was employed for all DFT calculations performed within the FLARE active learning framework and subsequent validation steps. All calculations used the generalized gradient approximation (GGA) exchange-correlation functional of Perdew-Burke-Ernzerhof (PBE)[57] and projector-augmented wave (PAW) pseudopotentials. Semi-core corrections of the pseudopotential and spin-polarization were not included for Au. A cutoff energy of 450 eV was employed, with an artificial Methfessel-Paxton temperature[58] of the electrons set at 0.2 eV for smearing near the Fermi-energy. Brillouin-zone sampling was done on a system-to-system basis, using **k**-point densities of 0.15 Å$^{-1}$ for periodic systems and Γ-point sampling for the NP systems.

### Production ML-MD simulations

ML-MD simulations were performed using a custom LAMMPS[59] pairstyle compiled for FLARE[31]. GPU acceleration was achieved with the Kokkos performance portability library[60], the performance of which for FLARE has been detailed elsewhere[31]. All simulations employed the Nosé-Hoover NVT ensemble with a timestep of 5 fs, which is appropriate for the mass of Au[39]. Velocities were randomly initialized for all simulations to a Gaussian distribution centered at 300 K. Each slab was built using the minimized lattice constant of Au, as predicted by the FLARE MLFF (4.16 Å) in the Atomic Simulation Environment[61]. To ensure that subsurface atoms were in registry with the bulk crystal structure, the bottom two layers of each slab were frozen, using the LAMMPS 'set_force' command to constrain the forces on these atoms to zero throughout the course of each simulation. Once each system was equilibrated over the course of 100 ps, dynamics were then observed at 300 K for a total of 10 ns, and thermodynamic and positional information was dumped every 1 ps. This dumping frequency was chosen specifically to allow for mechanistic study of each trajectory. All simulations at higher temperatures were initialized for 100 ps at 300 K, then the temperature was increased linearly at 20 K per ns until the desired temperature.

To explore the conditions that nucleate reconstruction, several modifications to each surface slab were also introduced. Point defects were included via random deletion of atoms in the top-most surface layer. Compressive and tensile strains were also introduced to both perfect and defective Au slabs using the LAMMPS 'change_box' command. All production ML-MD trajectories were analyzed with the OVITO software[41].

### Reporting summary

Further information on research design is available in the Nature Portfolio Reporting Summary linked to this article.

## Data availability

The data that support the findings of this study are available via the Materials Cloud Archive[62,63] and from the corresponding authors upon request.

## Code availability

An open-source implementation of the FLARE code is available at https://github.com/mir-group/flare and the snapshot release for the code used in this work is provided at https://github.com/mir-group/flare/releases/tag/1.3.0.

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

## Acknowledgements

We gratefully acknowledge Karsten Jacobsen for thoughtful discussions regarding interpretation of the simulation results and locality arguments made in the discussion. We thank Dr. Jin Soo (David) Lim and Dr. Jonathan Vandermause for helpful conversations at the outset of this project. We also thank Dr. Jenny Coulter for advice regarding implementation and analysis of the phonon dispersion results obtained via Phonopy and Phoebe codes. This work was primarily supported by the US Department of Energy, Office of Basic Energy Sciences Award No. DE-SC0022199 as well as by Robert Bosch LLC. C.J.O. was supported by the National Science Foundation Graduate Research Fellowship Program under Grant No. (DGE1745303). A.J. was supported by the NSF through the Harvard University Materials Research Science and Engineering Center Grant No. DMR-2011754. L.S. was supported by Integrated Mesoscale Architectures for Sustainable Catalysis (IMASC), an Energy Frontier Research Center funded by the US Department of Energy, Office of Science, Office of Basic Energy Sciences under Award No. DE-SC0012573. This research used resources of the National Energy Research Scientific Computing Center (NERSC), a DOE Office of Science User Facility supported by the Office of Science of the U.S. Department of Energy under Contract No. DE-AC02-05CH11231 using NERSC award BES-ERCAP0024206. Additional computational resources were provided by the FAS Division of Science Research Computing Group at Harvard University.

## Author contributions

C.J.O. initiated the study with L.S. and B.K., created the data set, compiled and analyzed the data, performed all model training, validation, ML-MD simulations, and post-processing, created all figures, and wrote the manuscript. Y.X. aided in FLARE implementation and calculations, and provided detailed feedback on the FLARE methods section. A.J. provided helpful advice regarding the GPU implementation of the FLARE code. L.S. advised components at the outset of this work. B.K. supervised all aspects of this work. All authors contributed to the revision of the manuscript.

## Competing interests

The authors declare no competing interests.
