## [Peer Review File · Nature Communications]

Low-index mesoscopic surface reconstructions of Au surfaces using Bayesian force fieldsREVIEWER COMMENTS

Reviewer #1 (Remarks to the Author):

The authors present a study of surface reconstruction using a machine-learning force field obtained via an active learning procedure. The particular novelty lies in the direct emergence of the collective behavior from the force field trajectory itself without any human bias in training or setup. This method is applied to facets and nanoparticles. It is particularly surprising to see that the comparably short cutoff of the model of about 6 Angstrom still allows for a description of the system that seems to be close to experimental information. The authors make predictions regarding the reconstruction dependence on strain which may be experimentally accessible. The work is well-written and easily accessible. The promise of the authors to publish raw data and model is commendable.

I see no issue with the manuscript and recommend publication as is, since the time and length scale covered and the depth of the analysis seems to warrant the venue.

Reviewer #2 (Remarks to the Author):

The manuscript "Stability, mechanisms and kinetics of emergence of Au surface reconstructions using Bayesian force fields" by Cameron J. Owen, Yu Xie, Anders Johansson, Lixin Sun, and Boris Kozinsky reports important insights on the nucleation of the surface reconstruction of Au and their time evolution under different effects that can influence it like stoichiometric, strain, and temperature.

The main points of criticism are as follows:

Fig. 1 is only mentioned in the text. I would suggest to describe and make more use of the figure in the text.

Similarly, to my previous comment, some figures (e.g. Fig. 2) are only properly described in the caption but not in the text. This also happens for other figures. A detailed description of the figures and their explanation should also be added to the text. This would considerably help the reader to follow the ideas of the manuscript.

Some figures are not mentioned in order (e.g. Fig. 4a before Fig. 2) which in my view is not beneficial for understanding.

The explanation concerning the table presented in Fig. 2b can be improved. The values on the x-axis correspond to the surface ML coverage of adatoms, but the way the table is schematically presented gives the reader the impression that this is related to the number of hcp environments (z scale) or even applied strain. The table should be edited so that the reader can clearly and directly identify the x- and y-axis. Additionally, I suggest adding a z-scale next to the table indicating what the values represent.

In Fig. 2a the meaning of the blue color BCC should be included in the caption of the image.

In the text the authors should indicate to the reader which part of the images they should pay attention to. This is essential for the understanding of the manuscript especially when referring to figures that contain many information. For this is extremely important first to describe the figures and subsequently explain them in detail. As an example, in section 1 (stoichiometric Au(111)) after describing the table the authors should indicate to the reader to first observe the first column of the table; and mention that the other columns refer to defective Au(111) which will be discussed later.

When referring to the SI ideally the authors should indicate which figures/sections the readers should

look at.

Fig. 3c should be explained in the text.

Could the full herringbone (Fig. 4) only be captured for a scale of 330000 atoms? Is the number of atoms the main difference when compared to previous simulations? It is not clear to me why the herringbone reconstruction could not be previously captured in the previous simulations.

The use of the same color for different purposes can lead to misinterpretation. For example, in the beginning of the manuscript red is used for hcp domains, and later on, it denotes atoms in the surface atomic layer. The color code is also confusing in Fig. 7

The discussion about Fig. 7 remains obscure to me and it is difficult to follow the ideas/explanations concerning this figure.

Reviewer #3 (Remarks to the Author):

The present manuscript presents a atomistic study of emerging mesoscopic structures on gold surfaces using a MLFF. It is obviously methodologically trailblazing and is very relevant contribution to the continued efforts to bridge the scales from surface science to model catalysis.

I have two objections that must be considered before publication.

1. Of the "four fundamental surface science questions addressed", the "Are short-range interatomic interactions sufficient to capture intricate large-scale reconstruction patterns observed in experiment?" is very weak

Obviously they are. There are textbook examples demonstrating this. The corresponding discussion subsection also uses a lot of words to say basically very little. It also gives the impression that the interactions described by a deep-learning model are limited by the extent of the local (but overlapping) descriptors, which is misleading

The relevant question would be "Are short-range interatomic interactions sufficient to model large-scale reconstruction patterns in quantitative agreement with experiment?". That might be the case in gold, but this is not properly addressed in the present study.

The study would loose little if this aspect was removed.

2. The precision of the underlying DFT calculations rely on a number of details. These play an important role for the Au surface reconstructions and have been discussed controversially in the literature. A starting point could be DOI: 10.1103/PhysRevB.82.161418

The authors don't address this aspect at all. There is no justification of the underlying DFT methodology or it's influence on the conclusions drawn.

Stability, mechanisms and kinetics of emergence of Au surface reconstructions using Bayesian force fields

Cameron J. Owen^{†,1} Yu Xie,² Anders Johansson,² Lixin Sun^{†,2} and Boris Kozinsky^{†2,3}

¹*Department of Chemistry and Chemical Biology,
Harvard University, Cambridge, Massachusetts 02138, United States*

²*John A. Paulson School of Engineering and Applied Sciences,
Harvard University, Cambridge, Massachusetts 02138, United States*

³*Robert Bosch LLC Research and Technology Center,
Watertown, Massachusetts 02472, United States*

RESPONSE TO THE REVIEWERS

We appreciate the time and effort that the editorial team and the three referees have invested in reviewing our manuscript. We address the comments and questions raised by each reviewer below. **The green text** refers to what was present in the old version of the manuscript, while **the blue text** indicates changes or additions to the revised manuscript. Importantly, we have included necessary statements regarding the choice of exchange-correlation functional, which properly place our work in the larger context of the reconstruction literature, as well as improved the readability with respect to the order and content of the figures. We hope that these modifications have improved the quality of the work, and again, appreciate your time in this effort.

REVIEWER 1

Reviewer: The authors present a study of surface reconstruction using a machine-learning force field obtained via an active learning procedure. The particular novelty lies in the direct emergence of the collective behavior from the force field trajectory itself without any human bias in training or setup. This method is applied to facets and nanoparticles. It is particularly surprising to see that the comparably short cutoff of the model of about 6 Angstrom still allows for a description of the system that seems to be close to experimental information. The authors make predictions regarding the reconstruction dependence on strain which may be experimentally accessible. The work is well-written and easily accessible. The promise of the authors to publish raw data and model is commendable.

I see no issue with the manuscript and recommend publication as is, since the time and length scale covered and the depth of the analysis seems to warrant the venue.

Author reply: We thank the reviewer for their strong support of the manuscript and for their time.

REVIEWER 2

Reviewer: The manuscript “Stability, mechanisms and kinetics of emergence of Au surface reconstructions using Bayesian force fields” by Cameron J. Owen, Yu Xie, Anders Johansson, Lixin Sun, and Boris Kozinsky reports important insights on the nucleation of the surface reconstruction of Au and their time evolution under different effects that can influence it like stoichiometric, strain, and temperature. The main points of criticism are as follows: Fig. 1 is only mentioned in the text. I would suggest to describe and make more use of the figure in the text. Similarly, to my previous comment, some figures (e.g. Fig. 2) are only properly described in the caption but not in the text. This also happens for other figures. A detailed description of the figures and their explanation should also be added to the text. This would considerably help the reader to follow the ideas of the manuscript. Some figures are not mentioned in order (e.g. Fig. 4a before Fig. 2) which in my view is not beneficial for understanding. The explanation concerning the table presented in Fig. 2b can be improved. The values on the x-axis correspond to the surface ML coverage of adatoms, but the way the table is schematically presented gives the reader the impression that this is related to the number of hcp environments (z scale) or even applied strain. The table should be edited so that the reader can clearly and directly identify the x- and y-axis. Additionally, I suggest adding a z-scale next to the table indicating what the values represent. In Fig. 2a the meaning of the blue color BCC should be included in the caption of the image. In the text the authors should indicate to the reader which part of the images they should pay attention to. This is essential for the understanding of the manuscript especially when referring to figures that contain many information. For this is extremely important first to describe the figures and subsequently explain them in detail. As an example, in section 1 (stoichiometric Au(111)) after describing the table the authors should indicate to the reader to first observe the first column of the table; and mention that the other columns refer to defective Au(111) which will be discussed later. When referring to the SI ideally the authors should indicate which figures/sections the readers should look at. Fig. 3c should be explained in the text. Could the full herringbone (Fig. 4) only be captured for a scale of 330000 atoms? Is the number of atoms the main difference when compared to previous simulations? It is

not clear to me why the herringbone reconstruction could not be previously captured in the previous simulations. The use of the same color for different purposes can lead to misinterpretation. For example, in the beginning of the manuscript red is used for hcp domains, and later on, it denotes atoms in the surface atomic layer. The color code is also confusing in Fig. 7 The discussion about Fig. 7 remains obscure to me and it is difficult to follow the ideas/explanations concerning this figure.

Author reply: We thank the reviewer for their suggestions regarding the interpretability and flow of the text. We address each comments from the reviewer below.

Reviewer: 1) Fig. 1 is only mentioned in the text. I would suggest to describe and make more use of the figure in the text.

Author reply: We have added several mentions of Fig. 1 to the text, specifically during the initial mention of the figure in the introduction, as well as in the Results summary section and subsequent section A.

Added:

As provided in Fig. 1, small atomic unit-cells of Au bulk, surfaces, and nanoparticles (left-most panel) are fed into the FLARE framework (middle-left panel) to efficiently trained a FLARE MLFF ‘on-the-fly’, from which unbiased ML-MD simulations can be performed (middle-right panel) to uncover atomistic understanding of surface reconstructions and their nucleation (right-most panel) directly from first principles.

Added:

The active learning procedure is represented schematically in Fig. 1, where atomic structures are input to the initially empty sparse Gaussian process model, which is trained iteratively via MD simulations.

Added:

The simulation procedure is described in detail in the Methods section and is also represented schematically in Fig. 1, where the trained FLARE model is used to describe various Au facets and nanoparticles under stimuli with unbiased MD to yield atomistic understanding.

Reviewer: 2) Similarly, to my previous comment, some figures (e.g. Fig. 2) are only properly described in the caption but not in the text. This also happens for other figures. A

detailed description of the figures and their explanation should also be added to the text. This would considerably help the reader to follow the ideas of the manuscript.

Author reply: We have attempted to clarify, where possible, each mention of figures within the main text to address this point. All figures are mentioned in sequential order, and we believe that the level of technical details in the figure references and captions are now appropriate.

Reviewer: 3) Some figures are not mentioned in order (e.g. Fig. 4a before Fig. 2) which in my view is not beneficial for understanding.

Author reply: The reference to Fig. 4a has now been removed, and the order in which figures are introduced in the text has been confirmed to be sequential.

Original:

The resulting ‘Herringbone’ periodicity ($22 \times \sqrt{3}$) has been experimentally imaged using scanning tunneling microscopy (STM), which can be found in References [2, 3] and is also provided here for the reader in Fig. 4(a).

Changed to:

The resulting ‘Herringbone’ periodicity ($22 \times \sqrt{3}$) has been experimentally imaged using scanning tunneling microscopy (STM), which can be found in References [2, 3].

Reviewer: 4) The explanation concerning the table presented in Fig. 2b can be improved. The values on the x-axis correspond to the surface ML coverage of adatoms, but the way the table is schematically presented gives the reader the impression that this is related to the number of hcp environments (z scale) or even applied strain. The table should be edited so that the reader can clearly and directly identify the x- and y-axis. Additionally, I suggest adding a z-scale next to the table indicating what the values represent.

Author reply: We have reformatted Fig. 2b by adding a z-scale and renaming the header of the x-axis to be more explicit. We believe these changes are sufficient for easy interpretation of the results in Fig. 2b.

Reviewer: 5) In Fig. 2a the meaning of the blue color BCC should be included in the caption of the image.

Author reply: We have included an explanation in the caption for the blue and white color assignments (the latter for environments with no symmetry).

Original:

(a) Final snapshot of the Au(111) surface after 10 ns of simulation time under 2% isotropic tensile strain. Atoms are colored using the Polyhedral Template Matching (PTM)[5] in Ovito [7], where green denotes *fcc* and red denotes *hcp* symmetry of a given atomic environment.

Changed to:

(a) Final snapshot of the Au(111) surface after 10 ns of simulation time under 2% isotropic tensile strain. Atoms are colored using the Polyhedral Template Matching (PTM)[5] in Ovito [7], where green denotes *fcc*, red denotes *hcp*, blue denotes *bcc*, and white denotes lack of symmetry of a given atomic environment.

Reviewer: 6) In the text the authors should indicate to the reader which part of the images they should pay attention to. This is essential for the understanding of the manuscript especially when referring to figures that contain many information. For this is extremely important first to describe the figures and subsequently explain them in detail. As an example, in section 1 (stoichiometric Au(111)) after describing the table the authors should indicate to the reader to first observe the first column of the table; and mention that the other columns refer to defective Au(111) which will be discussed later.

Author reply: We have added several statements which address this suggestion across all figures that are mentioned in the text. We point the readers explicitly to various parts of each figure, and agree that this increases the interpretation of the results.

Reviewer: 7) When referring to the SI ideally the authors should indicate which figures/sections the readers should look at.

Author reply: All mentions of the SI now include specific references to figures/sections.

Reviewer: 8) Fig. 3c should be explained in the text.

Author reply: Fig. 3c now has an extended description in Section A.4 of the Results, as provided below.

Added:

From these snapshots, clusters of atoms differing from the original *fcc* symmetry can be identified, which then begin to grow over a period of ~ 20 ps. Only once the growth has reached a critical surface area, do *hcp* domains begin to appear. We do not permit an extended focus on the study of nucleation phenomena here, but not that this method can be used to make such determinations, which may be the subject of future investigations.

Reviewer: 9) Could the full herringbone (Fig. 4) only be captured for a scale of 330000 atoms? Is the number of atoms the main difference when compared to previous simulations? It is not clear to me why the herringbone reconstruction could not be previously captured in the previous simulations.

Author reply: The Herringbone simulation can be captured at smaller scales using our method, as long as the periodicity of the surface supercell permits such a geometry (e.g. replicas of $22 \times \sqrt{3}$ unit cells). The number of atoms used in Fig. 4a was also chosen to demonstrate the attainable scale with our method, which allows for direct comparison of the simulated results to experimental observations like STM. Previous empirical methods also studied these surfaces with similar length-scales (comparable numbers of atoms), but due to the underlying difference in descriptions of atomic interactions, they were not able to capture the appearance of reconstruction from simulation initialized with the perfect facet. This question pertains more to the difference in the underlying functional forms that are used to describe the physics, versus the number of atoms used in the simulation.

Reviewer: 10) The use of the same color for different purposes can lead to misinterpretation. For example, in the beginning of the manuscript red is used for *hcp* domains, and later on, it denotes atoms in the surface atomic layer. The color code is also confusing in Fig. 7

Author reply: We have confirmed that the color scheme is in fact consistent across all figures *within* sections in the text, where red represents *hcp* domains as classified by the PTM method for the (111) surfaces, and is then used to represent surface atoms in the (110) and (100) studies. For clarity, we have included an explicit mention of the difference in

assignment in the text when transitioning between the (111) and (100) results. This is primarily a stylistic choice, and does not influence the interpretation of the results as all atomic color assignments are explicitly stated in the caption of each figure.

Added:

We make an explicit note here that the color scheme used for the analysis of each surface is unique, so the color assignments should be used across facets of different symmetry. For example, the *hcp* environments which are denoted using the color red and the atoms in Fig. 5 should not be interpreted in the same fashion, and these differences are made explicit in the caption of each figure.

Reviewer: 11) The discussion about Fig. 7 remains obscure to me and it is difficult to follow the ideas/explanations concerning this figure.

Author reply: We have clarified the discussion related to Fig. 7 by adding more references to the figures and explicit areas of focus within.

REVIEWER 3

Reviewer: The present manuscript presents a atomistic study of emerging mesoscopic structures on gold surfaces using a MLFF. It is obviously methodologically trailblazing and is very relevant contribution to the continued efforts to bridge the scales from surface science to model catalysis. I have two objections that must be considered before publication. 1. Of the "four fundamental surface science questions addressed", the "Are short-range interatomic interactions sufficient to capture intricate large-scale reconstruction patterns observed in experiment?" is very weak. Obviously they are. There are textbook examples demonstrating this. The corresponding discussion subsection also uses a lot of words to say basically very little. It also gives the impression that the interactions described by a deep-learning model are limited by the extent of the local (but overlapping) descriptors, which is misleading. The relevant question would be "Are short-range interatomic interactions sufficient to model large-scale reconstruction patterns in quantitative agreement with experiment?". That might be the case in gold, but this is not properly addressed in the present study. The study would loose little if this aspect was removed. 2. The precision of the underlying DFT calculations rely on a number of details. These play an important role for the Au surface reconstructions and have been discussed controversially in the literature. A starting point could be DOI: 10.1103/PhysRevB.82.161418 The authors don't address this aspect at all. There is no justification of the underlying DFT methodology or it's influence on the conclusions drawn.

Author reply: We thank the reviewer for their support of our work and valuable suggestions. We address each suggestion below. Before doing so, we would like to quickly point out that the method employed is not of the 'deep-learning' family, but rather an instance of Bayesian kernel regression, via the use of a sparse Gaussian process. This is critical in the interpretation of the conclusions, as the geometric descriptors centered on each atom only contains information of the local environment with a maximum distance of 6 Å, and there is no information propagation between the different environments, making this an interpretable strictly-local many-body additive energy model. In other different force field models, such as message-passing neural networks with multiple layers, such information propagation can occur and lead to explicit long-range many-

body interactions, but this is not the case in our model. We do not find the conclusion obvious, as the length-scale of the finite descriptor of the atomic environment and the length-scale of the reconstruction are on drastically different orders (6 Å for the descriptor versus ~ 60 Å for the length-scale of the reconstructions). We have added a short paragraph that addresses this in the introduction, and is provided below. This difficulty of constructing such a model is further corroborated by a wealth of literature that has not been able to simulate this reconstruction from the initially pristine facet, despite enormous effort over many years, as discussed in the introduction. We incorporate this understanding with the reviewer’s suggestions below.

Added:

This argument is similar to another critical component of the work completed here, in that not only are small approximates used to train a MLFF, but the descriptors used to represent atomic environments in the FLARE code are strictly local, where information is not propagated beyond the cutoff distance. Symmetry-breaking phenomena, such as charge density waves, which arise from delicate interactions among the various orbitals, are governed by long-range fundamental electronic structure features [1], which may be difficult to capture using strictly local representations. However, our non-trivial finding is that the emergence of such long-range patterns of reconstruction, due to strain and electronic effects, can in fact be described in quantitative agreement with experiment by a model that is able to sufficiently accurately capture only short-range quantum interactions.

Reviewer: 1) Of the "four fundamental surface science questions addressed", the 'Are short-range interatomic interactions sufficient to capture intricate large-scale reconstruction patterns observed in experiment?' is very weak. Obviously they are. There are textbook examples demonstrating this. The corresponding discussion subsection also uses a lot of words to say basically very little. It also gives the impression that the interactions described by a deep-learning model are limited by the extent of the local (but overlapping) descriptors, which is misleading. The relevant question would be "Are short-range interatomic interactions sufficient to model large-scale reconstruction patterns in quantitative agreement with experiment?". That might be the case in gold, but this is not properly addressed in the present study. The study would lose little if

this aspect was removed. The precision of the underlying DFT calculations rely on a number of details. These play an important role for the Au surface reconstructions and have been discussed controversially in the literature. A starting point could be DOI: 10.1103/PhysRevB.82.161418. The authors don't address this aspect at all. There is no justification of the underlying DFT methodology or it's influence on the conclusions drawn.

Author reply: We thank the reviewer for these important considerations. The first one (with respect to the 'obviousness' of the conclusions) was addressed above. As for the remaining points concerning the choice of the exchange correlation functional, we have now included a short summary of preceding methods that have tried to simulate these surface reconstructions on Au, with a particular emphasis on the choice of the exchange-correlation functional. We note that the provided reference establishes a stark disagreement between the use of PBE and all electron density functional theory for the accurate prediction of surface energies for the Pt(100) system, but does not make the same case for Au(100). Moreover, the provided reference considers a different issue altogether, which is the use of small periodicity 'approximates' to predict the surface energy of much larger ($N \times 5$) unit cells for the Pt and Au (100) reconstructions. This is not equivalent to the simulation task here, where we do not use small periodicities to predict mesoscopic properties, but rather we directly simulate the entire reconstruction with the resultant MLFF. Regardless, we have taken this opportunity to situate our work within the direct context of other MLFF practioners who have studied the resultant 'Herringbone' reconstruction (e.g. DeePMD using training labels at the PBE-level), obtaining agreement with experiment on various perturbations of the subsurface atomic layers, and strain induced changes in periodicity. However, we reiterate the observation that this previous work using the DeePMD potential did not directly simulate the transformation from pristine facet to reconstruction, and did not also consider the other low-index facets or nanoparticles.

Added:

However, we do find it apparent here to note that the work using the DeePMD potential took advantage of DFT training labels at the PBE level, which when used to study the resultant 'Herringbone' reconstruction (e.g. DeePMD using training labels at the

PBE-level), agreement was obtained with respect to experimental observations for various perturbations of the subsurface atomic layers, and strain induced changes in periodicity. There also exists a wealth of previous static DFT investigations, which when taken together conclude that the choice of exchange-correlation functional, albeit important, does not influence the stability assessments of the reconstructed facets relative to the pristine systems when using generalized gradient approximations like PBE [4, 6]. Issues may arise when using small periodicity ‘approximates’ (supercell representations, as defined in [4]) to predict the surface energy of the mesoscopic reconstructions as identified in [4] for Pt(100), but this is distinct from the simulation tasks presented here, since we simulate the entire length-scale required for the full surface reconstruction of each facet. Moreover, the work from [4] specifically notes that even the use of small periodicity approximates for the Au(100) at the GGA level with something like PBE give reasonable agreement to experimental observations.

OTHER CHANGES

- We also corrected a few typos and cleaned up syntax in the manuscript. We will release the model and data during the editorial revision stage following successful acceptance of this rebuttal, at which point we will provide the corresponding link to the Materials Cloud.

-
- [1] Marisol Alcántara Ortigoza, Maral Aminpour, and Talat S. Rahman. Friedel oscillations responsible for stacking fault of adatoms: The case of Mg(0001) and Be(0001). Phys. Rev. B, 91:115401, Mar 2015.
 - [2] Thais Chagas, Kai Mehlich, Abdus Samad, Catherine Grover, Daniela Dombrowski, Jiaqi Cai, Udo Schwingenschlögl, and Carsten Busse. Nucleation stage for the oriented growth of tantalum sulfide monolayers on au(111). The Journal of Physical Chemistry C, 127(11):5622–5630, 2023.
 - [3] Weiwei Gao, Thomas A. Baker, Ling Zhou, Dilini S. Pinnaduwaage, Efthimios Kaxiras, and Cynthia M. Friend. Chlorine adsorption on au(111): Chlorine overlayer or surface chloride? Journal of the American Chemical Society, 130(11):3560–3565, 2008.
 - [4] Paula Havu, Volker Blum, Ville Havu, Patrick Rinke, and Matthias Scheffler. Large-scale surface reconstruction energetics of pt(100) and au(100) by all-electron density functional theory. Phys. Rev. B, 82:161418, Oct 2010.
 - [5] Peter Mahler Larsen, Søren Schmidt, and Jakob Schiøtz. Robust structural identification via polyhedral template matching. Modelling and Simulation in Materials Science and Engineering, 24(5):055007, may 2016.
 - [6] Pai Li and Feng Ding. Origin of the herringbone reconstruction of au(111) surface at the atomic scale. Science Advances, 8(40), 2022.
 - [7] Alexander Stukowski. Visualization and analysis of atomistic simulation data with OVITO—the Open Visualization Tool. Modelling and simulation in materials science and engineering, 18(1), JAN 2010.

REVIEWERS' COMMENTS

Reviewer #2 (Remarks to the Author):

The authors have addressed my comments/suggestions and the manuscript can be accepted as it is.

Reviewer #3 (Remarks to the Author):

I must admit a certain annoyance with authors answer. In my opinion the authors answer a "fundamental surface science question" by now using even more words to say very little. But it's a somewhat non-falsifiable statement and I don't think continuing such a discussion here serves a purpose. There is enough positive aspects of the work to recommend publication.

The discussion of the previous "herringbone" work is satisfactory.